



# Combined Simulation and Optimization Framework for Irrigation Scheduling in Agriculture Fields

Mireia Fontanet[1,2,3], Daniel Fernàndez-Garcia[2,3], Gema Rodrigo[1], Francesc Ferrer[1], Josep Maria Villar[4]

[1]LabFerrer, Cervera, 25200, Spain
[2]Department of Civil and Environmental Engineering, Universitat Politècnica de Catalunya (UPC), Barcelona, 08034, Spain
[3]Associated Unit: Hydrogeology Group (UPC-CSIC)
[4]Department of Environmental and Soil Sciences, Universitat de Lleida (UdL), Lleida, 25003, Spain

*Correspondence to*: Mireia Fontanet (mireia@lab-ferrer.com)

**Abstract.** In the context of growing evidence of climate change and the fact that agriculture uses about 70% of all the water available for irrigation in semi-arid areas, there is an increasing probability of water scarcity scenarios. Water irrigation optimization is therefore one of the main goals of researchers and stakeholders involved in irrigated agriculture. Irrigation scheduling is often conducted based on simple water requirement calculations without accounting for the strong link between water movement in the root zone, soil-water-crop productivity and irrigation expenses. In this work, we present a combined simulation and optimization framework aimed at estimating irrigation parameters that maximize the crop net margin. The simulation component couples the movement of water in a variably saturated porous media driven by irrigation with crop water uptake and crop yields. The optimization component assures maximum gain with minimum cost of crop production during a growing season. An application of the method demonstrates that an optimal solution exists and substantially differs from traditional methods. In contrast to traditional methods, results show that the optimal irrigation scheduling solution prevents water logging and provides a more constant value of water content during the entire growing season within the root zone. As a result, in this case, the crop net margin cost exhibits a substantial increase with respect to the traditional method. The optimal irrigation scheduling solution is also shown to strongly depend on the particular soil hydraulic properties of the given field site.

## 1 Introduction

Agriculture is the largest consumer of freshwater and accounts for 70% of current human water use (FAO, 2011). The Food and Agriculture Organization of the United Nations (FAO) predicts that if current consumption patterns continue at the present rate, two-thirds of the world's population could be living in water-stressed countries by 2025. As a result, water will become scarce not only in arid areas but also in regions where precipitation is abundant (Pereira et al., 2002). Within this context, optimal irrigation water strategies are crucial for saving water while guaranteeing maximum crop yields in a near future.

Irrigation scheduling is the process used by irrigation system managers to determine both the correct moment and the required amount of water to irrigate fields while maximizing crop yields with the minimum amount of water applied. Properly defining


the moment of irrigation through water content (or pressure head) thresholds has been demonstrated to substantially increase the efficiency of agriculture fields (Dabach et al., 2013). However, irrigation water contains salts and fertilizers that often promotes soil salinization, which results in an increase in soil Electrical Conductivity ($EC$) and a reduction of crop productivity (Machado and Serralheiro, 2017). Thus, optimal irrigation strategy should be design to avoid soil salinization (Pereira et al.,
35  2007).

Irrigation scheduling strongly depends on water governance (Budds and Hinojosa, 2012; Hanjra et al., 2012) and directly controls irrigation system performance (Pitts et al., 1996; Sakthivadivel et al., 1993; Singh et al., 2006), water allocation (Dudley et al., 1971; Reca et al., 2001), and water accounting (Perry, 2011). Because of the importance of irrigation scheduling, a wide range of irrigation methodologies, such as soil water content and suction sensors installation, have been recently
developed and used in the field to guarantee crop productivity as well as to avoid soil salinization. We distinguish between methods based on crop water requirements, direct measurements of the plant water status or response to water stress, direct measurements of water content, and numerical modeling of flow and transport of salt concentrations in the vadose zone.

Water requirements determined from crop evapotranspiration calculations ($ET_c$) is the most widely used method for irrigation scheduling (Feng et al., 2007; Orgaz et al., 2006; Salim et al., 1970). $ET_c$ is typically estimated as the product of two
terms: the reference evapotranspiration ($ET_0$), generally defined for either clipped grass (Allen et al., 1998) or alfalfa (Wright and Jensen, 1978), and the specific crop coefficient ($K_c$), estimated from different tabulated information such as FAO56 Irrigation and Drainage Paper No. 56 (Allen et al., 1998). In this case, the daily or weekly volume of water evapotranspired is estimated and used to schedule the irrigation in the next days or weeks (Pereira et al., 2009; Sun et al., 2006; Thompson et al., 2007). This method does not provide the frequency and duration of irrigation (stakeholders do not know when to apply this
volume of water) and requires accurate estimations of weather conditions.

Plant based methods include direct measurements of the plant water status as well as a number of plant processes that are known to depend on water deficits (Jones, 2004). Different types of measurements can be used to determine plant water and salt stress for irrigation scheduling. This includes photosynthesis capacity (Flexas et al., 2004; de Lima et al., 2015; Ribas-carbo et al., 2006), stomatal conductance (Flexas et al., 2004; Jones, 1999), leaf water potential (Alberola et al., 2008; Girona
et al., 2006; Turner, 1990), and crop temperature (Bellvert et al., 2014; DeJonge et al., 2015; Kassie et al., 2018). In general, these methods only provide a direct measurement of the plant status without determining how much water is necessary. Moreover, these methods require sophisticated and calibrated devices (Jones, 2004).

Different sensors have been recently developed for the continuous measurement of water content, pressure head and $EC$ with high precision and temporal resolution (Ferrarezi et al., 2015; Hanson et al., 1977). With these measurements it is possible
to determine the Plant Available Water (PAW) (Denmead and Shaw, 1962) needed to estimate crop development and soil salinization for irrigation scheduling (Jones, 2004). The installation of these sensors is easy and fast. The location and depth of the sensor may substantially affect the estimation of PAW, which strongly depends on soil heterogeneity and root distribution (Fontanet et al., 2018). Several authors have studied the optimal irrigation strategy in different field areas and crop types using sensors (Hoppula and Salo, 2007; Phene and Howell, 1984; Thompson et al., 2007). These works provide the





threshold value for soil drying but results are case specific and cannot be directly extrapolated to other crop and fields with different soil types and weather conditions.

Several algorithms for optimizing irrigation can be found in the literature. For instance, Soentoro et al. (2018) optimized irrigation by determining the cropping patterns and planting areas through linear programming. Ortega et al. (2004) and Martínez-Romero et al. (2017) determined irrigation by maximizing the gross margin through genetic algorithms. Noory et al.

(2011) proposed a linear and a mixed-integer linear model for optimizing irrigation water allocation and multicrop planning that maximizes the total net benefit. These irrigation scheduling methods are based only on water requirements and do not properly represent the water movement in the vadose zone. Several authors have addressed the problem and made efforts to derive thresholds of irrigation for different soil types with different soil hydraulic properties based on the unsaturated hydraulic conductivity and potential plant water uptake for different root distributions (Srivastava and Yeh, 1991). Even though their

work remained theoretical, several authors (Collin et al., 2019; Gendron et al., 2018; Létourneau et al., 2015; Létourneau and Caron, 2019; Rekika et al., 2014) have later on derived the theoretical relationship and tested it in the field with success. Campbell, (1982) described at which soil water content fringe the crop is under optimal conditions. This soil water content fringe depends on soil hydraulic properties and it is shown to strongly vary in space (Feki et al., 2018). This indicates that it is important to consider soil water movement through the root zone during irrigation scheduling.

Numerical models constitute an efficient tool for assessing irrigation scheduling (Linker et al., 2016; Ma et al., 2015). Among the different models available, HYDRUS (Šimůnek et al., 2008, 2016) is often used to simulate water fluxes, root water uptake, root growth, and solute and heat transport in the vadose zone. Several researchers have used HYDRUS for simulating water content, soil suction and $EC$ to improve irrigation scheduling or to provide some information to stakeholder. For instance, Arbat et al. (2008) simulated soil suction with HYDRUS in order to demonstrate that this model is capable to

assess irrigation scheduling in a given field site. Siyal and Skaggs, (2009) and Skaggs et al. (2010) simulated soil moisture distribution patterns during drip irrigation. The water balance components were not analyzed in this study to ultimately define an optimal irrigation strategy. A different point of view was proposed by Twarakavi et al. (2009), who defined the field capacity point by simulating the drainage process with HYDRUS. These authors found that the field capacity point controls the irrigation schedule strategy but important variables such as transpiration were not considered. The role of transpiration was

latter on studied by Dabach et al. (2013), who analyzed different irrigations scheduling strategies through numerical simulations. In all these works, the assessment of irrigation scheduling disregards the strong link between the water balance components and the crop yield and economical profit.

In this work, we present a combined simulation and optimization framework aimed at obtaining the irrigation scheduling parameters that maximizes crop yield with minimum applied water while guaranteeing maximum net profit without soil

salinization. Importantly, in comparison with other methods, the optimal irrigation solution provided here fully couples the water movement in the root zone with irrigation expenses and profits obtained by the soil-water-crop productivity relationship under potential soil salinization conditions. The paper is organized as it follows. We first present the formulation of the optimization framework in section 2. Then, we applied the method to a specific field site in section 3. To do this, we set up a





numerical model to simulate flow and transport through the vadose zone at the field site. The model is first shown capable to

simulate water content data recorded at different depths over one year. After this, we optimize the irrigation problem and we discuss the results in section 4. The optimal irrigation solution is compared against traditional methods in this section. Finally, we also used the proposed framework to analyze the impact of soil properties on irrigation scheduling.

## 2 Combined Simulation – Optimization Framework

### 2.1 Optimization problem

The irrigation optimization problem is formally presented here. The framework permits to find the optimal control settings of an irrigated field that maximize the net profit obtained in a period of time $T$. Without loss of generality, we consider minimum fluctuations in the water price. That is to say that $T$ is a short period of time that refers for instance to the duration of a crop growing season. Larger periods of time can be considered by simply incorporating a discount rate in the water price. Let us consider that the schedule of irrigation depends on a vector of control variables $\mathbf{p}$ that characterize the irrigation rate $q_i(\mathbf{p}, t)$.

The components of this vector can include for instance the pressure head threshold ($h^*$), which indicates the degree to which the soil can dry before irrigation is applied, and the duration of irrigation ($\tau$) among others. The optimal irrigation scheduling strategy characterized by $\mathbf{p}$ Eq. (1) is defined here as the one that is most productive and sustainable in terms of minimum water applied, which is mathematically formulated as the problem of finding the set of control variables $\mathbf{p}$ that maximizes the crop Net Margin (NM) Eq. (2) cost  subject to operational and functional constraints. The NM cost is the revenue from crop

production, i.e., the Gain Margin (GM) Eq. (3) subtracted by the cost of crop production during an operational time $t = T$. The cost of crop production consists of two main parts: capital costs and operation costs. Infrastructure cost ($Capex$) Eq. (4) are the one-time expenses that usually incur during the purchase of land and equipment, i.e., expenses for bringing the irrigation field to an operable status. This includes the construction and installation of physical facilities such as the watering system, access roads, pipelines, drilling of wells and so on. Operation costs ($Opex$) Eq. (5) refers to the cost of specific activities

incurred during the crop field lifecycle, which include equipment maintenance, product transport and overheads. From this, the optimization problem is mathematically formulated as

$$\mathbf{p}_{opt} = \max_{\mathbf{p}} NM(\mathbf{p}, T), \tag{1}$$

where,
$$NM(\mathbf{p}, T) = GM(\mathbf{p}, T) - Opex(\mathbf{p}, T) - Capex, \tag{2}$$

$$GM(\mathbf{p}, T) = Y_a(\mathbf{p}, T)\, C_y, \tag{3}$$




$$Capex = C_m + C_c, \tag{4}$$

$$Opex(\mathbf{p}, T) = C_{Fix} + \int_0^T q_i(\mathbf{p}, t') (C_{wVar} + C_e) \, dt'. \tag{5}$$

The $GM$ [€·ha-1] is described here as the product between harvest price $C_y$ [€·ht-1] and the actual crop yield $Y_a$ [t·ha-1], $C_{Fix}$ [€·ha-1] is the water fixed cost, $C_{wVar}$ [€·m-3] is the water variable cost, $C_e$ [€·m-3] is the energy cost, $C_m$ [€·ha-1] is the

irrigation system maintenance cost, and $C_c$ [€·ha-1] is the capital cost.

In practice, irrigation control settings must satisfy also some operational constraints. The constraints can include limitations on irrigation parameters as well as limitations on leaching water quality (solute concentrations). These features are incorporated into the optimization problem by constraining the solution to practical limitations and requirements, formally written as

$$\mathbf{p_1} \leq \mathbf{p} \leq \mathbf{p_2} \tag{6}$$

$$g(c) \leq a \tag{7}$$

The first set of constraints Eq. (6) refers to practical issues such as those determined from the water system capacity installed. Here, we only set an upper and a lower bound of $\mathbf{p}$ to represent this. The other constraints Eq. (7) refer to solute concentrations, which can be used to limit for instance soil salinization. Note though that this methodology is not affected by the removal or the modification of the constraints included.

Solving this optimization problem results in the selection of optimal irrigation parameters. The solution will identify the

maximum profitable solution subject to sustainable and feasible constraints. However, the simulation of crop yields requires two simulation components, one that simulates the distribution of water in the vadose zone as a result of irrigation and environmental conditions, and another that relates soil water content and water availability with crop yields. These two components are discussed in the following sections.

## 2.2 Flow and transport model

The simulation of water availability requires a numerical model capable to predict the flow of water in the unsaturated zone. The unsaturated flow is typically described by Richards' equation (Richards, 1931) Eq. (8), which can be written as



$$\frac{\partial \theta}{\partial t} = \nabla \cdot [K(h)\nabla h + K(h)\nabla z] - S, \tag{8}$$

where $\theta$ [-] is the volumetric water content, $h$ [hPa] is the water pressure head, $t$ is time, $S$ [T-1] is the source-sink term (includes root water uptake), and $K$ [LT-1] is the unsaturated hydraulic conductivity. Solving this partial differential equation requires the knowledge of the Soil Water Retention Curve (SWRC), which is the relationship between the water content and

the pressure head, and the Hydraulic Conductivity Curve (HCC), which is the relationship between the hydraulic conductivity and the pressure head. These curves are characteristic for different types of soils.

The solution of Richards' equation requires also the knowledge of initial and boundary conditions. For our purposes, the boundary condition specified at the soil surface plays an important role as it defines the amount of water infiltrated into the soil during irrigation. The water flux across the soil surface depends on both external conditions such as irrigation water rates

$q_i(\mathbf{p}, t)$ and the water content conditions in the soil. The corresponding boundary condition should represent for instance that run-off or ponding effect occurs when an irrigation rate exceeds the infiltration capacity of the soil or the fact that evaporation cannot exceed the capacity of the soil to deliver enough water to the soil surface. Solving Eq. (8) subject to a system-dependent boundary condition that limits the surface flux by the following two conditions Eq. (9)-(10) is often used to incorporate these features (Feddes et al., 1974; Neuman et al., 1974),


$$\left|q(z_{top}, t)\right| \le |X(t)|, \tag{9}$$

$$h_d \le h \le h_s, \tag{10}$$

Where $q(z_{top}, t)$ is the water flux at the soil top surface, $z_{top}$ is the z-coordinate of the soil top surface, $X(t)$ is the prescribed maximum potential rate of infiltration or evaporation given by meteorological conditions [LT-1], and $h_d$ and $h_s$ refer to dry and saturated water pressure heads, respectively. During evaporation, $X(t) > 0$ and this value represents the maximum

evaporation rate $E_p$.

The numerical model should also describe the water extraction by plant roots. In this context, the root water uptake $S(h, h_\phi, z, t)$ Eq. (11) is typically determined by the product of the water and salinity stress function $\alpha(h, h_\phi)$, the root density distribution function $\beta(z, t)$ in the vertical direction, and the potential transpiration $T_p(t)$,

$$S(h, h_\phi, z, t) = \alpha(h, h_{\phi,}) \, \beta(z, t) \, T_p(t), \tag{11}$$

where $h_\phi$ is the osmotic pressure head. Several stress models can be found in the literature. Among them, the model presented

by Feddes et al. (1978) and van Genuchten (1987) are the most widely used. Essentially, the latter model considers a smooth monotonic function with maximum root uptake at saturated conditions, while the other represents a piecewise linear function



with maximum uptake within a pressure (saturation) interval. Importantly, only the formulation presented by Feddes considers a transpiration reduction near saturation. Assuming an isothermal system, the osmotic pressure head depends only on the solute concentrations of the chemical compounds in water, i.e., $h_\Phi = F(c_i)$. This relationship is often determined based on empirical

relationships.

The simulation of solute concentrations is needed for two reasons: to evaluate constraints in concentrations (prevent soil salinization) and to estimate osmotic pressures. Solute transport in the unsaturated zone is typically described by the advection-dispersion equation Eq. (12), which is written as,

$$\frac{\partial (R\theta c_i)}{\partial t} = -\nabla \cdot (\boldsymbol{q} c_i) + \nabla \cdot (\theta \mathbf{D} \nabla c_i) + f_i, \quad i = 1, \dots, N_s \tag{12}$$

where $c_i$ is the solute concentration of the i-th chemical component, $\boldsymbol{q}$ is the Darcy flux, $\mathbf{D}$ is the hydrodynamic dispersion

tensor, $N_s$ is the number of chemical components considered, and $f_i$ is the concentration source-sink term.

**2.3 Soil – water – crop productivity relationship**

An estimation of the crop yield is required to evaluate the revenue from crop production during optimization. Crop productivity models can be complex as they include the interaction between genetics, physiology and environmental conditions such as water content. To facilitate the estimation procedure, one often considers an empirical model that relates crop water needs to

crop yields. The model presented by Stewart et al. (1977) Eq. (13) has been widely accepted and recommended by the Food and Agriculture Organization of the United Nations (FAO) (Doorenbos and Pruitt, 1975, hereafter FAO24). This simplified crop yield model represents the seasonal pattern of crop water needs by different growing stages (four in the case of maize). The potential crop yield is penalized at each growth stage depending on the deficit of water, which is estimated by the relative discrepancy between potential (maximum) and actual evapotranspiration. The model determines that


$$Y_a = Y_p \prod_{k=1}^{N_y} \left( 1 - K_{y_k} \left( 1 - \left( \frac{ET_a}{ET_c} \right)_k \right) \right) \tag{13}$$

where $Y_p$ [t·ha-1] is the potential crop Yield for the total growing season, $k$ is the growing stage index, $N_y$ is the number of growing stages, and $K_{y_k}$ [-] is the crop yield response factor associated with the k-th growing stage. $ET_a$ and $ET_c$ are the actual and potential accumulated crop evapotranspiration in each growing stage k [mm]. Note that the crop is under stress conditions

when $ET_a / ET_c < 1$.

At each growing stage, according to Allen et al. (1998) the potential evapotranspiration $ET_c$ Eq. (14) is estimated by the crop coefficient $K_c$ and the reference evapotranspiration $ET_0$. The latter can be calculated by the Penman-Monteith equation (Allen et al., 1998), which is a function of the input daily mean temperature, the wind speed, the relative humidity and the solar radiation.



$$ET_c = K_c \, ET_0 \,, \tag{14}$$

The $ET_a$ Eq. (15) is estimated from the simulation of flow in the vadose zone, which gives the evaporation at the soil surface and the water uptake by plant roots $S(h, h_\Phi, z, t)$. From this, the $ET_a$ of the k-th growing stage taking place in the time interval $(t_k, t_{k+1})$ can be determined by

$$ET_a = \int_{t_k}^{t_{k+1}} q(z_{top}, t') I(q(z_{top}, t') > 0) dt' + \int_{t_k}^{t_{k+1}} \int_{z_{top}-L_R}^{z_{top}} S(h, h_\Phi, z, t') dz \, dt', \tag{15}$$

where $I(q > 0)$ is an indicator function that is equal to one during evaporation ($q > 0$) and zero otherwise, $z_{top}$ is the z-
coordinate of the soil top surface, and $L_R$ is the vertical length of the root zone.

## 3 Field Application

### 3.1 Site description

In this section we illustrate the applicability of the method in a real field setting. The study area considers a 25 ha commercial farm located in Foradada, Spain (Fig.1). The field is divided into 23 irrigation sectors which are fully covered by sprinkles.
The irrigation rate is 6.5 L·m-2·h-1. Two sectors can be irrigated at the same time during 24 h·d-1. The soil texture can be classified as a Silty Clay Loam (USDA soil taxonomy) with 28% Clay, 58.4% Silt and 13.6% Sand. Every year two different crops are grown: The first crop is usually canola and the growing season extends between winter and spring, during which the soil is wetted by rainfall events. The second crop is maize which is grown during the summer and autumn when soil is irrigated. Irrigation scheduling is needed to maximize productivity.

A field campaign was conducted aiming to measure soil water dynamics and soil hydraulic properties. For this, soil water content sensors were installed and undisturbed soil core samples were collected. Two EC-5 volumetric water content sensors (METER Group, Pullman, WA, USA) were installed at 10 and 20 cm depth in a representative location (Fig.1). Sensor data was collected every 5 minutes with an accuracy of ±0.03 cm3·cm-3 (Campbell and Devices, 1986). Two undisturbed soil samples were taken nearby the EC-5 sensors using a stainless-steel ring of 250 cm3 capacity. These soil samples were used to
measure the SWRC and HCC with high precision (± 1.5 hPa) and over a wide range of pressures. This was achieved by combining the HYPROP, the WP4c, and the KSat devices (METER Group, Pullman, WA, USA). Whereas the HYPROP device is capable to measure SWRC and HCC, WP4c can complement SWRC in the dry region. The KSat system does the same for HCC. A comparison of approaches has been reported by Schelle et al. (2013). These authors demonstrated that this combined method shows less noise than the other traditional methods. The  van Genuchten–Mualem SWRC model was used
to fit experimental data using the HYPROP Fit software (METER Group, Pullman, WA, USA). Figure 2 shows SWRC and HCC measured by HYPROP, WP4c and KSat. Note that these devices provide SWRC and HCC with high resolution. SWRC





describes a soil with a quite high water content retention capacity and a low air enter potential and slope. This is shown by the shape parameters α and n with values of 0.0678 and 1.186, respectively. HCC is characterized by a low $K_s$ value with 12 cm·d-1, indicating slow wetting front movement during irrigation.

### 3.2 Model setup

We use the HYDRUS-1D software package (Šimůnek et al., 2008, 2016) for simulating the one-dimensional movement of water and solute transport in variably-saturated porous media at the field site. This code solves Richards' equation to simulate water flow in the unsaturated zone and the advection-dispersion equation to simulate solute transport using numerical methods based on the Galerkin finite element method. We consider a 60 cm vertical soil profile representative of the Foradada field site. The domain was discretized into 101 segment elements. The column represents the movement of water through the soil profile associated with the water content sensors and core samples. A system-dependent boundary condition was imposed at the soil top surface according to Eq. (9)-(10). Since the water table is far below, a free drainage boundary condition was imposed at the column bottom, i.e., $q = -K(h)$. Considering the soil type and the soil water content measurements, initial conditions were set to $\theta = 0.25$. A multiplicative model was used to represent the water and salinity stress function, i.e., $\alpha(h, h_\Phi) = \alpha(h)\alpha(h_\Phi)$, where $\alpha(h)$ and $\alpha(h_\Phi)$ are the water and salinity stress functions, respectively. In this case, we have considered the Feddes et al. (1978) Eq. (16) model to represent the water stress function,

$$\alpha(h) = \begin{cases} \dfrac{h - h_4}{h_3 - h_4} & h_3 > h > h_4 \\ 1 & h_2 \geq h \geq h_3 \\ \dfrac{h - h_1}{h_2 - h_1} & h_1 > h > h_2 \\ 0 & h \leq h_4 \ or \ h \geq h_1 \end{cases}, \tag{16}$$

which describes that the plant suffers water stress outside the pressure head range $(h_2, h_3)$. The water stress reduction decreases linearly from those pressure points and gets to a minimum $(\alpha = 0)$ below or above $h_4$ and $h_1$. Another function used is the salinity stress function which is defined using the threshold-slope salinity stress reduction function (Maas and Hoffman, 1977) Eq. (17),

$$\alpha(h_\Phi) = \begin{cases} 1, & a \leq h_\Phi \leq 0 \\ 1 + b(h_\phi - a), & a > h_\Phi > -\dfrac{1}{b} \\ 0, & h_\Phi \leq a - \dfrac{1}{b} \end{cases}, \tag{17}$$

Here, the salinity threshold value $a$ quantifies the minimum osmotic head above which root water uptake occurs without reduction, and the slope b determines the fractional root water uptake decline per unit increase in salinity below the threshold. The parameters adopted to define these stress functions were chosen from the HYDRUS internal database and are summarized





in Table 1. The transport model is simplified to simulate only one representative chemical component, i.e., Electrical Conductivity $EC$, which is assumed to behave as a conservative species (non-reactive). We neglect salt precipitation and dissolution processes. The osmotic pressure is assumed to be proportional to $EC$. Considering that $h_\phi$ and $EC$ are expressed here in the same units, we essentially have that $EC = h_\phi$ (Simunek and Sejna, 2014).

An initial estimation of model properties (e.g., SWRC and HCC) was known from core sample measurements. The model parameters were then calibrated to reproduce the recorded soil moisture data obtained at the field site. The calibrated parameters were not significantly different from the initial estimation. Table 2 summarizes the measured and calibrated parameters. Parameters modified in the calibration process were $\alpha$ and $n$. The simulation considered 240 days, from February 9th 2017 until October 31st 2017. Two different crops growing at different periods of time were accounted for during the

simulation; canola from February, 9th 2017 to May, 31st 2017, and maize from June, 1st 2017 to October, 31st 2017. Meteorological parameters were downloaded from the nearest available weather station to compute $ET_0$, which was then converted into daily $ET_c$ values with the $K_c$ coefficients shown in Table 3. The potential evaporation and transpiration values needed in the root water uptake model, Eq. (9) - (11), were calculated by partitioning $ET_c$ into potential evaporation $E_p$ and transpiration $T_p$ based on the Canopy Cover (Raes et al., 2010), which determines that $ET_c = \beta E_p + (1 - \beta)T_p$, being $\beta$ the

soil cover fraction. Figure 3 compares simulation results with soil moisture field measurements obtained at two different depths. Simulations are in good agreement with soil moisture data. Table 4 shows several goodness-of-fit statistics, such as Willmott index (Willmott, 1981), calculated for both depths.

### 3.3 Optimal irrigation scheduling problem setup

We applied the simulation-optimization framework presented in section 2 to the Foradada field site to estimate optimal

irrigation scheduling parameters during a growing season where $ET_c$ demand was the highest one from years from 2008 to 2018. Thus, in this specific case, the most unfavorable weather conditions were considered, even though the methodology allows for other weather conditions as well. The methods and parameters used here were directly adopted from the previous calibrated model. However, weather conditions considered the growing season with the highest $ET_c$ values estimated between 2008 and 2017 to represent dry conditions without rainfall events. The corresponding $K_c$ coefficients are presented in Table 3.

The year with more water demand was 2016. Initial conditions were $\theta = 0.25$ and $EC = 0.6$ dS/m, which represents an average water content of the field site without salinity problems. Simulations considered the entire growing season of maize, from June 15th (sowing) to November 11th (harvesting). This crop is cultivated during the dry season; thus, irrigation should be applied in order to ensure crop development. The rest of the year, rainfall events maintains soil under wet conditions and irrigation is not necessary. As a consequence, it is not necessary to schedule any irrigation during this period of time. Data necessary to

evaluate the crop net margin cost $NM$ is summarized in Table 5, mostly provided by the Aigües Segarra Garrigues (ASG) company.

Two control irrigation parameters were used to characterize water irrigation rates, i.e., the pressure head threshold $h^*$ observed




at a control point and the duration of irrigation $\tau$. The control point is located at the vertical midpoint of the maximum maize root length (20 cm below soil surface). It is considered the depth where the maize main roots are located. The pressure head at

the control point triggers irrigation when $h < h^*$ (when the soil is dry). At this moment, the model applies water at a constant irrigation rate $q_i$ for an irrigation time $\tau$. Since the watering capacity is fixed by the type of irrigation equipment installed, the irrigation rate is not considered to be a control parameter in this case ($q_i$ is set constant to 6.5 L·m-2·h-1). Figure 4 presents an illustrative sketch of the irrigation. The salt concentration in the irrigated water is assumed to have electrical conductivity values of 0.4 dS/m. The optimization is constraint to fulfil that $EC < 3.4$ dS/m (below this threshold maize is not under salinity

stress).

A large number of algorithms can be used to maximize the crop net margin cost function $NM$ with constraints. Here, we chose to maximize $NM$ over a given range by brute force, which simply consists in computing the function's value at each point of the parameter space to find the global maximum. This can be inefficient in practical applications but provides detail insights about irrigation scheduling as well as the full shape of the $NM$ cost function, which is the objective here. To do this, the

parameter space $(\tau, h^*)$ was discretized into a $4 \times 10$ regular mesh, where $\tau$ ranges between 1 and 4 hours and the threshold pressure head $h^*$ varies between -100 KPa and -10 KPa.

In order to analyze the performance of the method, we compared the optimal irrigation results obtained with our proposed framework with those given by a traditional irrigation method. The traditional irrigation scheduling method is based on water requirements and consists in irrigating as much water as that evapotranspirated in the previous week. To this end, the farmer

must devise an irrigation calendar. Thus, to simulate the traditional method, the weekly $ET_c$ value is calculated and this volume of water is applied over the next week. This amount of water is uniformly distributed over the next week. All other parameters are kept the same. The relative difference between the Net Margin obtained with the traditional method ($NM_{trad}$) and the optimal one is computed as follows,

$$\Delta_r NM = \frac{NM - NM_{trad}}{NM}. \tag{18}$$

## 305   4 Simulation - Optimization Results

In this section, we present the simulation-optimization results of the Foradada irrigation scheduling problem. Figure 5a shows a map of the net margin $NM$ function obtained from all the irrigation strategies simulated as a function of $h^*$ and $\tau$. A clear $NM$ maximum value of 2791 €·ha-1 can be seen in this figure for $\tau = 1$ h and $h^*$= -40 kPa. This optimal irrigation strategy represents a short but moderately frequent irrigation, i.e., moderate $h^*$ value, which results from balancing the gain margin

$GM$ with the operational expenses $Opex$. The corresponding maps of $GM$ and $Opex$ are also depicted in Fig. 5b and Fig. 5c, respectively. The maximum gain requires a more frequent irrigation with $\tau = 1$ h and $h^*$= -25 kPa (more irrigated water) but the operational expenses significantly increase in this region. Thus, even though this is the most productive strategy, $Opex$





penalizes economically $GM$ and, consequently, $NM$ decreases. Thus, although this irrigation strategy could be the most productive in crop yield terms, the required volume of water and expenses related can substantially affect the optimal irrigation

strategy. On the contrary, $Opex$ is minimum when the frequency of irrigation is very small, implying that less water is used for irrigation. Applying the optimal irrigation strategy, the volume of water applied is 470 mm. Figure 5d shows the relative difference between the traditional method and the optimal solution obtained with the proposed methodology. Results show that the proposed method can increase the net margin by 7%. We also note that the $NM$ function can give smaller values than the traditional one when either irrigations strategies apply a large volume of water (increasing $Opex$) or when $h^*$ is too small

(decrease in $GM$). This situation is far away from the optimal irrigation strategy and thereby the simulation-optimization method seems mandatory in routine field applications.

A more profound understanding of the difference between the traditional irrigation scheme and the optimal irrigation method ($\tau = 1$ h and $h^* = -40$ kPa) can be seen from Figure 6, which compares the temporal evolution of the water content resulting from both methods at 4 different soil depths during the growing season. The optimal irrigation strategy applies water for 1

hour during several days and stops when $h^*$ approaches -40 kPa. The traditional method applies water every day (the total of volume of water applied is 478 mm). The main difference is that the optimal solution provides a more uniform variation of the water content over the entire season, i.e., from 10 to 40 cm depth water content oscillates between 0.20 and 0.32 cm3·cm-3, compared with the traditional method, which fluctuates between 0.16 and 0.34. Moreover, the optimal method is capable to increase the water content globally in the root zone, in particular at 10 cm depth, whereas the traditional method stresses the

system during periods with higher $ET_c$ demand by letting the water content to decrease up to 0.16 cm3·cm-3. This water stress affects crop productivity because water content conditions are not optimal. We also note that when the volume of water applied is less than 5 mm·d-1, the wetting front does not arrive to the deepest soil parts, compromising root water uptake. This way, even though the traditional method applies the total volume of water evapotranspirated during the growing season, this is seen not enough to maintain constant the water content during the growing season. In contrast, optimal irrigation method is based

on soil water status and how water moves through the root zone. For this reason, applying the optimal irrigation method can be guaranteed that crop is under optimal conditions. Thus, irrigation scheduling optimization algorithms must be focused on which strategy produces higher economic benefit considering the most optimal interaction between crop and soil water conditions.

To better understand the results, Figure 7 describes the impact that each irrigation strategy produces on $T_a$, $E_a$ and soil $EC$.

The optimal strategy is highlighted in these figures with a black circle. Figure 7a presents the simulated transpiration values obtained as a function of $h^*$ and $\tau$. Remarkably, the optimal irrigation strategy is not the one that produces more transpiration but lies within the plant water stress region, i.e., $\alpha(h^*)$ is smaller but close to 1. This means that the gain in crop productivity obtained for $\alpha = 1$ does not compensate the expenses associated with the increase in irrigated water. Results also show a decrease in transpiration when irrigation applies water during several hours. This is caused by the saturation of the top horizon

of the soil (see Fig.8).

Figure 7b plots the corresponding evaporation values. According to Philip (1956) and Ritchie (1972), we distinguish between





two evaporation regions: Range I represents an energy-limited evaporation process where the soil surface is wetted by irrigation and water evaporates from a thin soil surface layer; Range II represents a falling-rate evaporation process that occurs when water content flows from the soil layer below. Results show that the optimal irrigation strategy is energy-limited because the

Foradada soil has a high-water content retention capacity.

Figure 7c presents the simulated $EC$ values at the end of the season as a function of $h^*$ and $\tau$. They are the average from the four observation points inserted at different depths. Even though none of the irrigation strategies exhibit salinity stress (above threshold) and increase in $EC$ is seen in all simulations. Consequently, salts are accumulated through the root zone presenting a maximum at the end of the season. This is more pronounced for small $h^*$ values. This increase could significantly affect the

crop productivity in the following seasons and reflects the importance of rainfall events in wet periods (winter and spring). This volume of water should be responsible for flushing out the salts accumulated during dry periods. Our simulated scenarios considered only dry conditions without rainfall events and thereby this flushing mechanism could not be seen.

Figure 8 displays the wetting front resulting from different irrigation events for 4 different irrigation strategies. One can observe that when water is applied during several hours (i.e., more that indicated by the optimal irrigation strategy), the top horizon of

the soil becomes almost saturated and thereby water uptake is negligible in this region (pressure head overpasses the $h_2$ threshold). This prolonged saturation of the top soil horizon due to poor drainage inhibits crop productivity. This is in turn reflected by smaller $T_a$ values in Figure 7a.

In order to evaluate the impact that soil hydraulic properties have on the optimal irrigation strategy, we also solved the optimization-simulation problem considering a different soil type. The chosen soil hydraulic properties have been downloaded

from the Rosetta database (Schaap et al., 2001) available from HYDRUS 1D. Table 2 shows the soil hydraulic parameters of a Loamy Sand soil. In this specific case, $\alpha$ and $n$ parameters represent a soil with less soil moisture retention capacity than the Foradada soil, and $K_s$ is substantially higher than the soil previously studied ($K_s = 350$ cm·d-1). Taking these observations into account, a lower soil moisture retention with a faster wetting front movement is expected. Figure 9 shows the map of the $NM$, $GM$, and $Opex$ function depending on $h^*$ and $\tau$. The linear-like shape of these functions are strikingly different from

those of Foradada, which exhibited a strong nonlinear behavior close to the optimal value. Consequently, results show that in this case $h^*$ is mostly controlling $NM$ and $\tau$ has little effect. An optimal irrigation strategy is found for $\tau = 1$ h and $h^* = -10$ kPa. The maximum $NM$ value is 2719 €·ha-1, which is slightly smaller than that of the Foradada soil of 2791 €·ha-1. This irrigation strategy represents a short but very frequent irrigation. We conclude then that the optimal irrigation strategy can drastically change from one soil type to another.

The physical process by which the $NM$ function associated with a Loamy Sand soil is now mostly controlled by the pressure head threshold $h^*$ and not $\tau$ can be seen from the dependence of $T_a$, $E_a$, and $EC$ with $h^*$ shown in Fig. 9. Note that in all cases these results are substantially different than those of the Foradada soil (Fig. 7). In this case, results indicate that the irrigation water can easily infiltrate and redistribute through the entire root zone, following a falling-rate evaporation process. In contrast to our previous results, a more permeable soil leads to an optimal irrigation strategy (black circle) that favors maximum

transpiration rates with zero plant water stress. The optimal strategy also inhibits waterlogging and provides minimum





salinization compared to other irrigation strategies obtained with different pressure head thresholds (Fig. 9f). Again, this can be explained by an effective percolation of water through the root zone, which gives good internal drainage.

## 5 Conclusions

Irrigation scheduling in agriculture is crucial for saving water while guaranteeing maximum crop yields in arid regions as well as in future areas affected by climate change and water scarcity. However, irrigation scheduling is typically conducted based on simple water requirement calculations without accounting for the strong link between water movement in the root zone, crop yields and irrigation expenses. In this work, we have presented a combined simulation and optimization framework aimed at estimating irrigation parameters that maximize the crop net margin cost subject to operational and functional constraints. The simulation component couples the movement of water in a variably saturated porous media driven by irrigation with plant water uptake and crop yields. The optimization component assures maximum gain with minimum cost of crop production during a growing season.

An application of the method was presented in the Foradada irrigation field test site, where soil hydraulic parameters represented a soil with high water content capacity and slow wetting front percolation. The method was demonstrated to yield optimum irrigation parameters at the site (irrigation duration of 1 hour and a frequency determined by a pressure head threshold of -40 kPa). These parameters were substantially different from those estimated with traditional water requirement methods based on previous evapotranspiration values. Results have shown that even though the volume of water evapotranspired during the growing season is fully replaced by the traditional method, the way to schedule irrigation does not guarantee that water content values vary within an optimal water content fringe. Prolonged saturation of the top soil horizon is often shown to occur with the traditional method. On the contrary, the optimal irrigation scheduling solution prevents waterlogging and provides a more constant value of water content during the entire growing season within the root zone. As a result, the crop net margin cost exhibited an increase with respect to the traditional method by a factor of 7%. The optimal irrigation solution is also demonstrated to properly balance the crop gain margin and operational expenses. Results have shown that even though some strategies can be more productive, irrigation expenses counterbalance the economic benefit ultimately leading to a compromise between them. At this stage, we highlight the practical advantages of the method proposed compared to traditional water requirement methods: (i) agriculture stakeholders can obtain better crop gain margins to achieve the same crop productivity; and (ii) irrigation scheduling parameters can be known prior to the start of the growing season. However, the method requires a more important soil characterization of the field site and the installation of sensors to measure (at least) pressure head potentials.

The impact of soil hydraulic properties has been also analyzed by assuming another soil type (Loamy Sand soil). Soil hydraulic parameters described a soil with less water retention capacity than the Foradada soil and easier wetting front percolation. Results have shown that the optimal solution strongly depends on the type of soil. In this case, the frequency of irrigation was much larger given by a pressure head threshold of -10 kPa.



Finally, our results indicate that the irrigation optimization algorithms that will be developed in the next future should account for the water movement trough the root zone instead on water requirements estimations only. Moreover, irrigation scheduling
should be based on soil pressure head status to guarantee optimal crop performance during the growing season. In order to achieve this, an accurate soil hydraulic characterization is crucial as well as the installation of  pressure head sensors to monitor the pressure head status during the season.

**Data availability**

All data are available from the corresponding author upon request.

**Author contributions**

MF and DFG contributed to design and implementation of the research, to the analysis of the results and to the writing of the paper while GR, FF and JMV evaluated results obtained.

**Competing interests**

The authors declare that they have no conflict of interest.

**Acknowledgments**

We thank to our colleagues from Aigües Segarra Garrigues (ASG) company for sharing some of the data necessary to conduct
this work and also to Doctorats Industrials to fund the project where this work is involved.

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

**Figures**

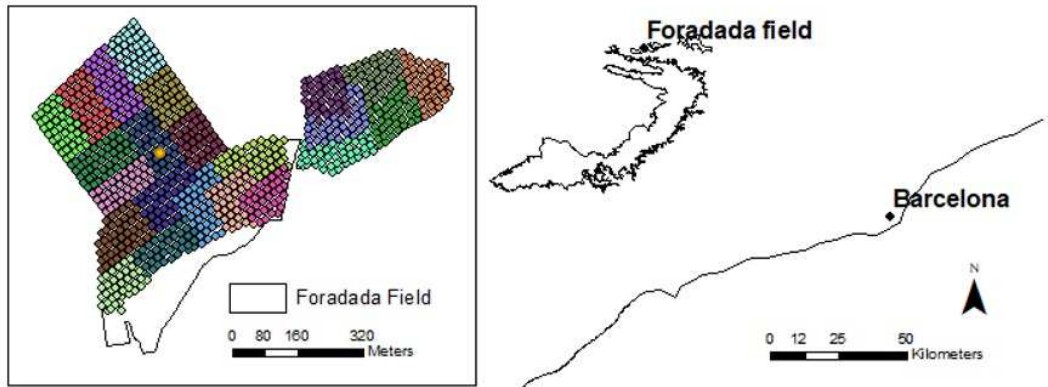

**Figure 1: The Foradada field is located within the Segarra Garrigues system (ASG) canal. Solid irrigation sprinkler systems installed**
**and distributed in 23 sectors are represented in different colors. Water content monitoring station and sampling is represented in**
**yellow.**

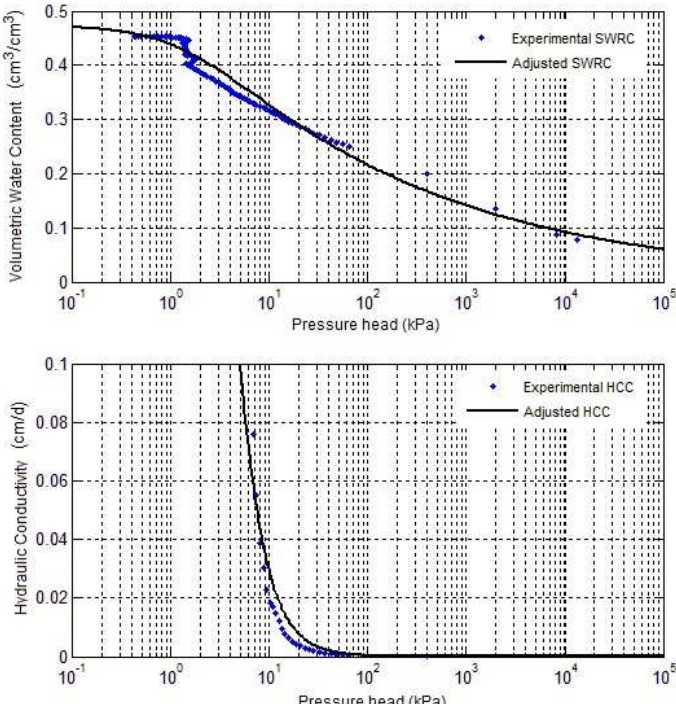

**Figure 2: Soil Water retention Curve (SWRC) and Hydraulic Conductivity Curve (HCC) measured by HYPROP, WP4c and KSat systems and fitted by van Genuchten – Mualem model.**





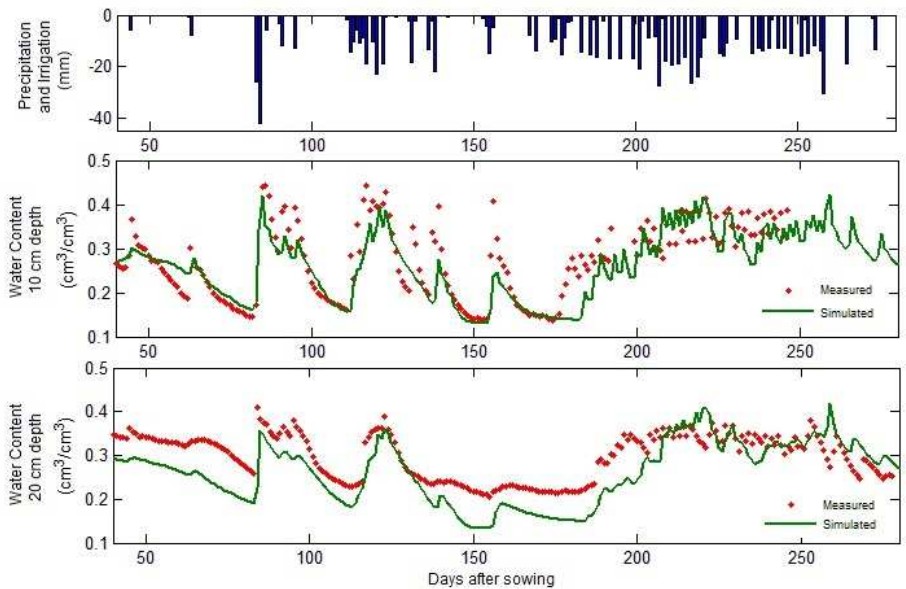


**Figure 3: Comparison between daily soil moisture field measurements and the soil moisture output from the validation model at 10 and 20 cm depth.**

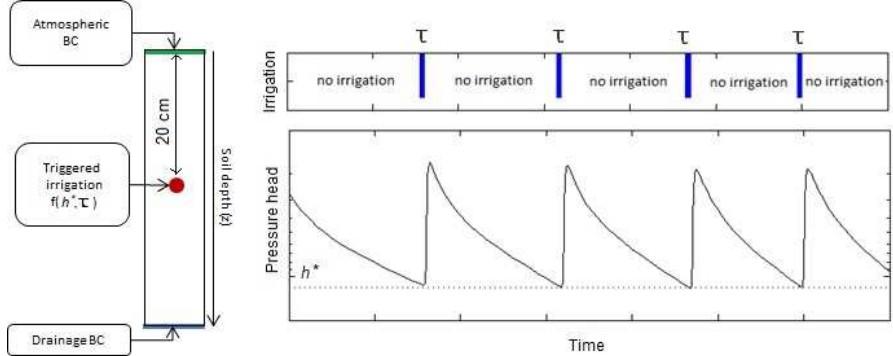

**Figure 4: On the right, boundary conditions imposed in the model where triggered irrigation is a function of $h^*$ and $\tau$. On the left, a**

**synthetic case about how the model triggers the irrigation, when $h^*$ and $\tau$ are defined.**

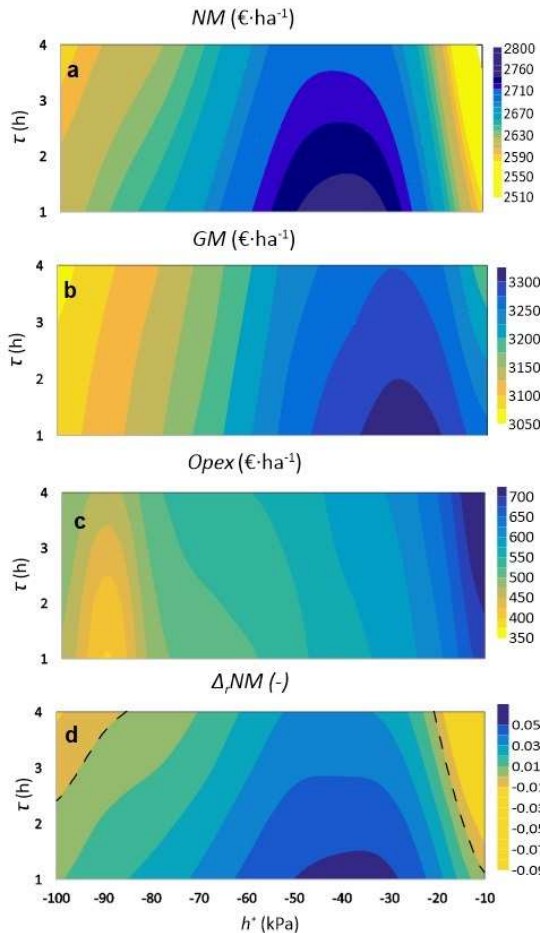

**Figure 5: a) Net Margin ($NM$) Foradada soil results and objective functions elements, being b) $GM$, Gain Margin; c) $Opex$, Operational costs and d) $\Delta_r NM$, fractional difference between $NM$ and $NM_{trad}$. Dash line in $\Delta_r NM$ map represents when the relative increment is zero.**

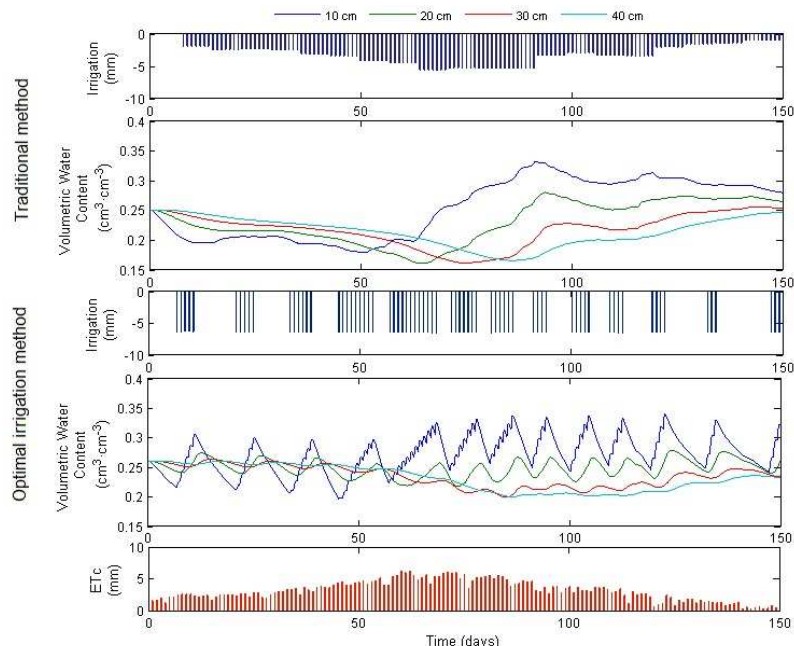


**Figure 6: Irrigation scheduling from both strategies, where irrigation, volumetric water content dynamics at different depths and the potential evapotranspiration demand ($ET_c$) are plotted.**



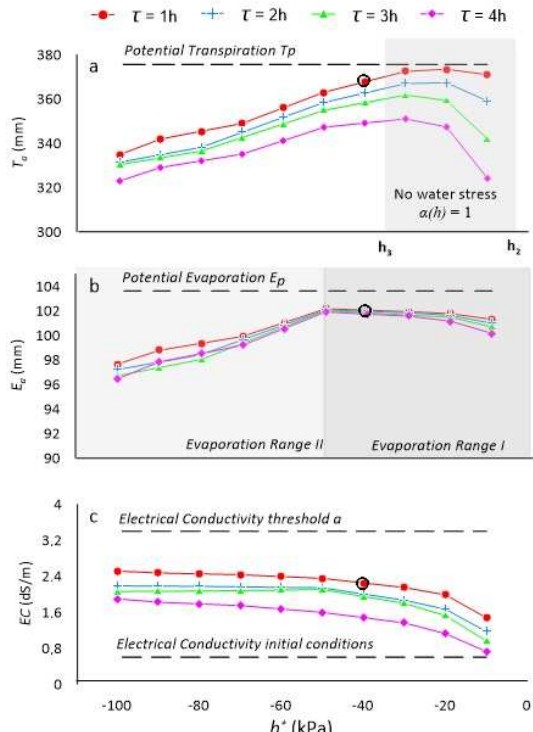

**Figure 7: a) Actual Transpiration ($T_a$), b) actual Evapotranspiration ($E_a$) and c) Electrical Conductivity at the root zone ($EC$)**

**resulting from Foradada soil showing all the irrigation strategies simulated. Circles show the strategy who provides maximum Net Margin ($NM$).**


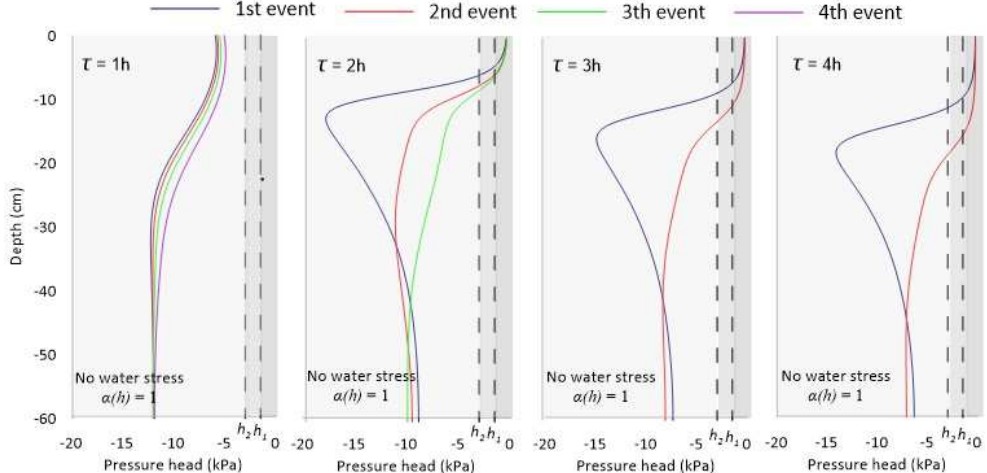

**Figure 8: Wetting patterns when irrigation strategy is fixed at $\tau$ = 1, 2, 3, 4 h and $h^*$ = 10 kPa. Some water stress function parameters ($h_1$ and $h_2$) are plotted indicating when transpiration decreases as a consequence of water logging.**

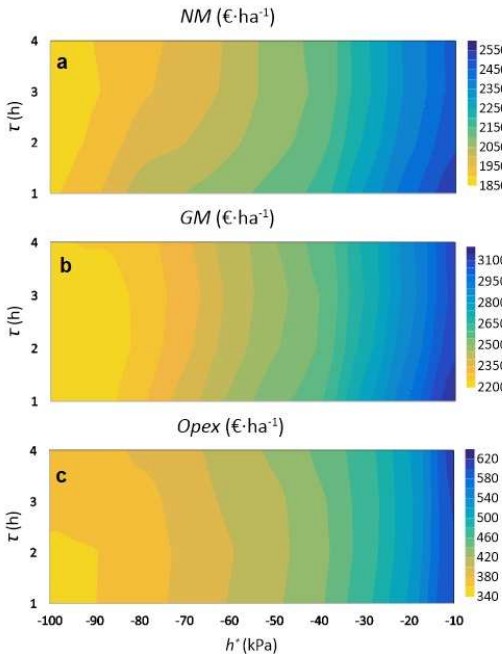


**Figure 9: a) Net Margin ($NM$) Loamy sand soil results and objective functions elements, being b) $GM$, Gain Margin and c) $Opex$, Operational costs, d) Actual Transpiration ($T_a$), e) actual Evapotranspiration ($E_a$) and f) Electrical Conductivity at the root zone ($EC$). Circles show the strategy who provides maximum Net Margin ($NM$).**





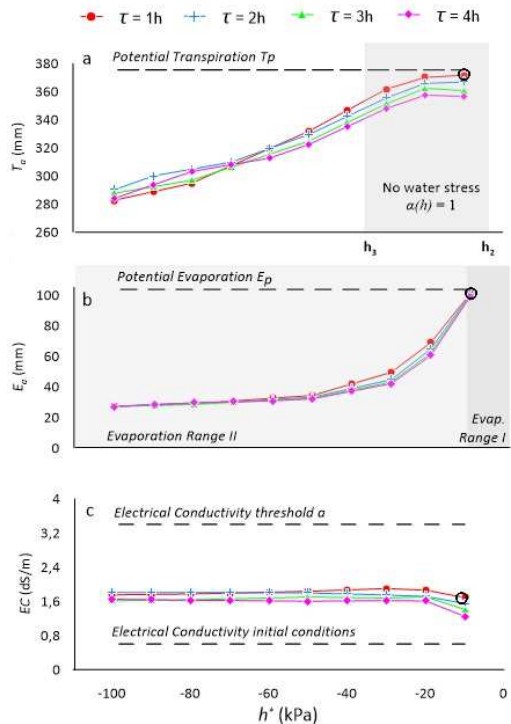

**Figure 10: a) Actual Transpiration ($T_a$), b) actual Evapotranspiration ($E_a$) and c) Electrical Conductivity at the root zone ($EC$) resulting from Loamy Sand soil showing all the irrigation strategies simulated. Circles show the strategy who provides maximum Net Margin ($NM$).**

**Tables**

| | Feddes Function (water stress) | | Mass and Hoffman (salinity stress) | | |
|---|---|---|---|---|---|
| $h_1$ | 1.5 | kPa | a | 3.4 | dS/m |
| $h_2$ | 3.0 | kPa | b | 6 | - |
| $h_3$ | 32.5 – 600.0 | kPa | | | |
| $h_4$ | 800.0 | kPa | | | |

**Table 1. Water and salinity stress function parameters.**



| Soil hydraulic parameters | Foradada soil | | Theoretical soil | Units |
|---|---|---|---|---|
| | Measured | Calibrated | Loamy Sand | |
| $\theta_r$ | 0.012 | 0.012 | 0.05 | cm3·cm-3 |
| $\theta_s$ | 0.473 | 0.473 | 0.41 | cm3·cm-3 |
| $\alpha$ | 0.0421 | 0.0678 | 0.124 | 1·h-1 |
| $n$ | 1.157 | 1.186 | 2.28 | - |
| $K_s$ | 12 | 12 | 350 | cm·d- |
| $I$ | 0.5 | 0.5 | 0.5 | - |

**Table 2. Foradada and Loamy Sand Soil hydraulic parameters used in simulations. $\theta_r$, is the residual volumetric water content; $\theta_s$, is the saturated volumetric water content; $\alpha$, $n$ and $i$ the are shape parameters, and $K_s$ is the saturated hydraulic conductivity.**

| Stage | I | II | III | IV |
|---|---|---|---|---|
| Kc1 | 0.2 | 0.7 | 1.15 | 0.2 |
| Kc2 | 0.3 | 0.3 – 1.1 | 1.1 | 1.10 – 0.55 |
| Ky | 0.35 | 1.05 | 0.4 | 0.2 |

**Table 3. Canola crop coefficient (Kc1), maize crop coefficients (Kc2) applied for ETc, where Stage I represents the initial period; Stage II is crop development; Stage III mid-season; Stage IV is late season. Crop yield response factor (Ky) used for actual Yield estimations, where Stage I is vegetative period, Stage II is flowering period, Stage III the yield formation, and Stage IV ripening.**

| Observation Point | RMSE | Willomtt Index | R2 |
|---|---|---|---|
| θ 10 cm Depth | 0.12 | 0.89 | 0.61 |
| θ 20 cm Depth | 0.08 | 0.96 | 0.60 |

**Table 4. Statistical index calculated with observed and simulated water content values.**







| Parameter | Value | Units | Reference |
|---|---|---|---|
| $Y_p$ | 19.5 | t·ha-1 | Martínez-Romero et al. (2017) |
| $C_y$ | 171.8 | €·t-1 | www.mapama.gob.es |
| $C_m$ | No data available | €·ha-1 | - |
| $C_c$ | No data available | €·ha-1 | - |
| $C_{Fix}$ | 115.35 | €·ha-1 | Aigües Segarra Garrigues |
| $C_{wVar}$ | 0.1003 | €·m-3 | Aigües Segarra Garrigues |
| $C_e$ | No data available | €·m-3 | - |
| $q_i$ | 6.5 | l·m-2·h-1 | Aigües Segarra Garrigues |

**Table 5. Parameters necessary to apply Stewart and Net Margin ($NM$) equations. $Y_p$ is the potential crop yield; $C_y$ is the harvest price; $C_m$, is the maintenance cost; $C_c$, are the capital costs; $C_{Fix}$, is the fix water cost; $C_{wVar}$, is the variable water cost; $C_e$, is the energy cost; $q_i$, is the irrigation rate.**



