# Peer review of "Combined Simulation and Optimization Framework for Irrigation Scheduling in Agriculture Fields"

_Hydrology and Earth System Sciences, 2020_

## Referee Comment (RC1) · Anonymous Referee #1 · 18 Apr 2020

This study proposed a method to locate optimal irrigation schedules considering the soil water movement. The research question is interesting, but I am not sure if it is relevant within the scope of HESS. In addition, I found the complexity of modeling practices could be consistent and optimization methods could be improved so that the results could be more reliable and practical. For instance, the crop yield model looks too simple (only a function of ET), compared to that of the soil water model (HYDRUS-1D). There are more comprehensive crop models such as DSSAT and EPIC. The brute force method used when trying to locate the optimal scheduling could be fine if the authors wanted to see the relationship between two factors or objective functions, but it is not an efficient way to explore the multi-dimensional parameter space. Such a limitation did not allow the authors to explicitly investigate the trade-off between the objective functions and develop a Pareto front in the study. Here are my specific comments:

Lines 49 to 50: I do not agree with this statement. The ET based method can provide information on irrigation water application timing when it is combined with soil water content accounting.

Equation 13: This model accounts only for the impacts of ET or soil water content on crop growth, and I think this is too simple compared to the complexity of using the soil water simulation model, HYDRUS-1D.

Line 221: Please briefly describe what these devices for.

Lines 256 to 257: Does this mean that the differences between them are not "statistically" significant? Please clarify it.

Line 266: I am not sure if we can say this. Please try to justify the evaluation using literature.

Lines 271 to 272: Please describe the weather conditions in detail.

Line 272: Please justify such selection of weather condition in terms of the reliability and applicability of the results. I think it is worth adding other weather conditions (e.g., most favorable and average) and comparing the efficiency of the proposed method.

Lines 292 to 293: I do not think the brute force sampling strategy can locate the global optimum.

Lines 293 to 294: Please provide examples of showing the detail insights about irrigation scheduling.

Lines 298 to 301: I do not think this is a "realistic" traditional irrigation scheduling method, which may determine daily (rather than weekly) irrigation timing and amount based on daily (rather than weekly) weather conditions.

HESSD
Line 306: I expected to see a plot showing the trade-off between the objective functions (or a Pareto front), but Figure 5 does not show it.

Line 318 ("proposed method can increase the net margin by 7%"): Please describe how the amount of water irrigated and the corresponding cost can be improved (or reduced) by implementing the proposed method.

Lines 400 to 401: Considering the amount of uncertainty in the analysis and its results, I am not sure the 7% increase of the net margin is significant. Please try to quantify uncertainty of this analysis, as there are many assumptions and simplifications made in the analysis and modeling.

Lines 404 to 408: Considering these shortcomings of this method, I am not sure if agriculture stakeholders can use this method in practice. I wonder how the authors are going to make this tool available to the stakeholders.

Figure 3: The model underestimated soil water content at the 20 cm depth, which may lead to the overestimation of irrigation water.

Figure 6: The optimal scheduling requires to turn on and off the irrigation pump and system frequently, which may lead to increase in operation and maintenance costs. I am wondering if such additional potential costs can be considered in the optimization framework.

Table 4: RMSE values, 0.12 and 0.08 look substantial when considering the fact that the amount of available water content is around 0.35 from Figure 3. 0.12 and 0.08 correspond to 33% to 23%.

Table 4: How about the overall bias?

---

## Referee Comment (RC2) · Anonymous Referee #2 · 10 May 2020

**General comments**

The authors present an interesting and generally well-written manuscript about optimizing irrigation scheduling by combining 1-dimensional simulation of soil water movement and crop yield optimization. The research question is relevant. The methods used for simulation are well established. The results are interesting and promising for further research building upon the described methods.

**Specific comments**

Line 59: The publication of Hanson et al. (1977) might be too old as a reference to 'recent developments'.

Line 74: You write "several authors" – are there more sources than Srivastava and Yeh (1991)?

Line 109: Please provide the units of $q_i$ at an earlier point than line 210.

Line 147: The unit of $h$ is not consistent with the rest of Richards' equation. Usually, the pressure head is given in units of length. Throughout the manuscript, please try to better distinguish 'pressure' from 'pressure heads'.

Line 148: L is used as symbol for 'length unit' as well as for 'liter' (as in line 210, 287) in the manuscript. Please try to remove this ambiguity.

Line 256: Can you briefly indicate the method used for calibration?

Line 261: What is the distance between the study site and the nearest available weather station?

Line 335/336: 'Guarantee' is a strong word – maybe use a weaker one. I agree that the "optimal irrigation method" seems to ensure better conditions than the "traditional method".

Line 401: The word 'factor' is misleading – maybe better write "increase with respect to the traditional method by 7%". Moreover: Are 7% significant, considering the uncertainties involved?

**Technical corrections**

Line 33: promote

Line 34: designed

Line 77: Campbell (1982)

Line 83: stakeholders

Line 84/85: capable of assessing

Line 85: Siyal and Skaggs (2009)

Line 90: later

Line 90: irrigation scheduling

Line 97: as follows

Line 98: apply

Line 99/100: capable of simulating

Line 102: use

Line 128: Units of $C_y$ should be [EUR.t-1].

Line 169: the models presented

Line 289: constrained

Line 328/329: capable of increasing

Line 353: an increase

Line 376: It should be Fig. 10.

Line 381: It should be a panel of Fig. 10.

Line 414: through the root zone instead of

Line 482: Reference seems to be incomplete/broken.

Table 2: Usually, the unit of $\alpha$ is [m-1] or [cm-1]. Units of $K_s$ should be [cm.d-1]. Please be consistent with the symbol for the third shape parameter $i/I$.

Table 4: Willmott

Table 5: For consistency, maybe better use 'L' as symbol for 'liter'.

---

## Referee Comment (RC3) · Anonymous Referee #3 · 25 May 2020

The paper suggests a procedure for identifying the optimal irrigation schedule that maximize the net margin of the crop production. Irrigation scheduling is defined by

i) a threshold of the soil matric potential observed at a given depth (e.g. 20 cm in the sample case study);

ii) event irrigation depth (or irrigation duration with a predefined irrigation rate) .

The procedure exploits Hydrus-1D as soil water and solute transport model. Plan transpiration is modelled as fraction of the potential transpiration. The fraction is computed accounting for water and salinity stress.

[Figure]

The potential transpiration and evaporation are computed as fractions of the crop evapotranspiration under standard conditions (ETc), accounting for the soil canopy cover.

ETc is computed according to FAO-56 single crop coefficient approach.

Crop yield is assumed to be proportional to the ratio of the actual ET to ETc.

*General Comments*

Tabulated FAO 56 crop coefficients were proposed as a simple approach for assessing crop water requirements. The application of the single crop coefficient approach for estimating the crop evapotranspiration under standard conditions and the crop yield is too simplistic and not suited for the proposed optimization. By taking the tabulated crop coefficient in the proposed procedure is equivalent to assume that the crop is a stationary system, where phenology and water requirements are simply identified by the calendar days rather than the result of the crop response to the environmental conditions.

Similarly, Eq. 13 was proposed by FAO papers as a simple empirical equation for estimating crop yield.

However, crop biomass and yield development depend on the transpiration rate rather than on the evapotranspiration.

Even the most simple and conceptual agro-hydrological model, such as AquaCrop (which does not rely on the numerical solution of the Richards Equations) provides a comprehensive description of the crop dynamics and crop yield development and, thus, allows optimizing the irrigation scheduling accounting for the crop response to environmental stresses.

Indeed, an optimization procedure should consider that environmental stresses do not affect the yield uniformly across the entire growing cycle.

Overall, it is not clear the motivation of this study. How should this procedure be applied

from an operational perspective? The optimization procedure seems to be designed for running in batch mode, i.e. it can be used to identify the optimal irrigation schedule for a reference climatic condition, but it cannot be used to adapt the irrigation schedule to the actual environmental conditions, in real-time.

*Specific comments*

Line 50 – The crop coefficient is designed for assessing crop water requirements and not for irrigation scheduling. The estimated crop water requirements should be then used for designing the irrigation scheduling.

Section 3.2 Model setup: Root depth is assumed to be constant in time, while it is highly variable in time, especially for crops like maize. A soil depth of 60 cm with free drainage as bottom boundary conditions does not seems to be realistic. Moreover, this seems even more improbable with crops like maize. The impact of the initial conditions can be high.

Lines 325 – The irrigation strategy presented as traditional does not seem to be realistic.

---

## Author Comment (AC1) · 20 Jul 2020

Ref: hess-2020-146 Title: Combined Simulation and Optimization Framework for Irrigation Scheduling in Agriculture Fields Journal: Hydrology and Earth System Science

Response to Reviewer #1 Comments ####Authors' response in Blue####

Review on hess-2020-146 "Combined Simulation and Optimization Framework for Irrigation Scheduling in Agriculture Fields"

We are grateful for the time and effort that the reviewers spent on the manuscript. Our response to the reviewers is attach to this document. We believe that our responses

and the revisions made to the manuscript fully address the issues raised by the review. These revisions have helped clarify some aspects of our work and improve its interpretation.

GENERAL COMMENTS

This study proposed a method to locate optimal irrigation schedules considering the soil water movement. The research question is interesting, but I am not sure if it is relevant within the scope of HESS. In addition, I found the complexity of modeling practices could be consistent and optimization methods could be improved so that the results could be more reliable and practical. For instance, the crop yield model looks too simple (only a function of ET), compared to that of the soil water model (HYDRUS1D). There are more comprehensive crop models such as DSSAT and EPIC. The brute force method used when trying to locate the optimal scheduling could be fine if the authors wanted to see the relationship between two factors or objective functions, but it is not an efficient way to explore the multi-dimensional parameter space. Such a limitation did not allow the authors to explicitly investigate the trade-off between the objective functions and develop a Pareto front in the study.

We know that Stewart (1977) model is simple but we decided to use this model because it has been widely accepted and recommended by FAO and used by several authors in recent years (Domínguez et al., 2012; Irmak et al., 2016; Martínez-Romero et al., 2017; Saadi et al., 2015). Note though that the ET values used by this model are determined from HYDRUS and are therefore consistent with soil water movement, salts and crop stress factors. This makes the application of this model quite more complex an accurate than used in practice. Nevertheless, it is important to point out that the paper presents a general methodology to optimize irrigation scheduling in agricultural fields. Therefore, how to model each compartment is not that important. We rather focus on the interplay between them. We added some clarifications specifying that several crop yield model can be applied instead Stewart (1977) model (Line 189):

"The model presented by Stewart et al. (1977) Eq. (13) has been widely accepted and recommended by the Food and Agriculture Organization of the United Nations (FAO) (Doorenbos and Pruitt, 1975, hereafter FAO24). In addition, it has been recently used by several authors (Domínguez et al., 2012; Irmak et al., 2016; Martínez-Romero et al., 2017; Saadi et al., 2015). Note though that methodology proposed is not limited by this model and can also be used with other soil – water – crop productivity models if needed."

We agree that the brute force method used to locate the optimal scheduling is not efficient. We actually state this in the manuscript in Line 318. We preferred to use brute force in this case in order to explore in detail the objective function as a function of the irrigation parameters. Of course, the methodology and the results presented do not depend on the algorithm to minimize the objective function.

The text in the manuscript where all of this is explained reads as it follows (Line 317):

A large number of algorithms can be used to maximize the crop net margin cost function NM with constraints. Here, we chose to maximize NM over a given range by brute force, which simply consists in computing the function's value at each point of the parameter space to find the global maximum. This can be inefficient in practical applications but provides detail insights about irrigation scheduling as well as the full shape of the NM cost function, which is the objective here. To do this, the parameter space ($\tau$,$\hat{h}^*$) was discretized into a $4\times10$ regular mesh, where $\tau$ ranges between 1 and 4 hours and the threshold pressure head $\hat{h}^*$ varies between -100 KPa and -10 KPa.

SPECIFIC COMMENTS:

Lines 49 to 50: I do not agree with this statement. The ET based method can provide information on irrigation water application timing when it is combined with soil water content accounting.

Agreed. We modified the sentence as it follows (Line 49):

"If water requirements are not combined with soil moisture sensors measurements, they do not provide the frequency and duration of irrigation (stakeholders do not know when to apply this volume of water) and requires accurate estimations of weather conditions."

Equation 13: This model accounts only for the impacts of ET or soil water content on crop growth, and I think this is too simple compared to the complexity of using the soil water simulation model, HYDRUS-1D.

Stewart (1977) model is based on ET and a crop response factor. We assume that it is a simple model but we specified in "General comments" section, that several authors applied this model recently and it is recommended by FAO. However, even this model is simple, input variables necessary to apply the model (ET values) are extracted from a HYDRUS simulation who contemplates soil water content patterns and salts concentration through the root zone. Based on FAO recommendation, the authors content that these values are representative. Therefore, if used as input variables in Stewart (1977) model crop yield results must be a good approximation.

Line 221: Please briefly describe what these devices for.

we explained what these devices for in the following sentence, please, let us know if you need more details (Line 229).

"Whereas the HYPROP device is capable to measure SWRC and HCC, WP4c can complement SWRC in the dry region. The KSat system does the same for HCC. A comparison of approaches has been reported by Schelle et al. (2013)."

Lines 256 to 257: Does this mean that the differences between them are not "statistically" significant? Please clarify it.

Sorry, the use of the word "significantly" in this sentence is confusing since it seems that we want to express "statistically significant". This is not what we meant. It is a simple

appreciation of the results, pointing to the fact that they do not substantially deviate from the initial estimate. We have substituted the word "significantly" by "substantially" to avoid this confusion. (Line 275)

Line 266: I am not sure if we can say this. Please try to justify the evaluation using literature.

We are not completely sure what is the reviewer asking here but we agree that the fitting is not that good at depth 20 cm during the first 200 days after sowing compared to the simulations obtained at depth 10 cm. We have changed the text in the manuscript to acknowledge this:

We modified the sentence as follows (Line 287):

"Figure 3 compares simulation results with soil moisture field measurements obtained at two different depths. Simulations are in good agreement with soil moisture data, except for a relatively small underestimation of the water content measured at depth 20 cm by a factor of about 1.15 during the first 200 days after sowing"

Lines 271 to 272: Please describe the weather conditions in detail.

We described the weather conditions as follows (Line 300):

"The year with more water demand was 2016 with an atmosphere demand of 478 mm and a total rainfall of 80 mm. During this period of time, the maximum and minimum temperatures were 39°C and 21°C, respectively."

Line 272: Please justify such selection of weather condition in terms of the reliability and applicability of the results. I think it is worth adding other weather conditions (e.g., most favorable and average) and comparing the efficiency of the proposed method.

Firstly, we want to clarify that the methodology proposed in this work can be applied with any weather conditions, from the most unfavorable conditions until the most favorable ones. In order to prove that the methodology will work under unfavorable conditions, we decided to simulate this particular case with the most unfavorable conditions because we think that the crop is more susceptible to be under water stress conditions than a year with low atmosphere demand. In this case, we assume that it will be more complicated to solve the simulation – optimization problem with a realistic result when atmosphere demand is high. The reason is because it will be more difficult to maintain soil under optimal soil moisture conditions.

Lines 292 to 293: I do not think the brute force sampling strategy can locate the global optimum.

The brute force explores the entire parameter space and therefore, by definition, can detect local and global maximum values at the expenses of CPU times. It is true though that the exploration requires defining discrete points where the objective function is evaluated and therefore the exact global optimum can slightly deviate from our results. We see though that the objective function is quite smooth and this smoothness is larger than the sampling frequency. This gives confidence to the results provided.

Lines 293 to 294: Please provide examples of showing the detail insights about irrigation scheduling.

The "detail insights" about irrigation scheduling are already provided in section 4 of the present manuscript. In this section, we provide the maps of the objective function as a function of the parameters. From these maps, we give guidelines for improving irrigation.

Lines 298 to 301: I do not think this is a "realistic" traditional irrigation scheduling method, which may determine daily (rather than weekly) irrigation timing and amount based on daily (rather than weekly) weather conditions.

Our experience in the Segarra-Garrigues agricultural fields in Spain indicates that it is more convenient defining weakly water requirements. The reason is that agricultures must then reprogram the irrigation controller once per week and not daily, which is too

time consuming and annoying for them.

Line 306: I expected to see a plot showing the trade-off between the objective functions (or a Pareto front), but Figure 5 does not show it.

The main goal of this figure is to represent NM, GM... of each simulation, providing a global perspective bout how all irrigation strategies affect the system. In fact, we define here the optimal irrigation strategy, but we also describe useful information that must be considered, such as, the relationship between GM and Opex.

Line 318 ("proposed method can increase the net margin by 7%"): Please describe how the amount of water irrigated and the corresponding cost can be improved (or reduced) by implementing the proposed method.

We added the information required (Line 343):

"Results show that the proposed method can increase the net margin by 7%, decreasing by 6% the total amount of water applied at the end of the campaign, and reducing by 5% the costs associated by irrigation."

Lines 400 to 401: Considering the amount of uncertainty in the analysis and its results, I am not sure the 7% increase of the net margin is significant. Please try to quantify uncertainty of this analysis, as there are many assumptions and simplifications made in the analysis and modeling.

This is a synthetic case study designed to illustrate the method proposed. As such, the approach is deterministic. We think that a stochastic analysis of irrigation scheduling to account for the uncertainty in the spatial variability of the soil attributes is out of the scope of the present manuscript, whose objective is to present a general framework for optimizing irrigation. We do not discard to consider a stochastic framework in the future but this would require a full paper in itself.

Lines 404 to 408: Considering these shortcomings of this method, I am not sure if agriculture stakeholders can use this method in practice. I wonder how the authors are

going to make this tool available to the stakeholders.

Thanks. In section 5 we have rewrite the sentence to clarify how a stakeholder must implement the method. The text reads as it follows (Line 436).

"In order to implement the method some measurements are required. Firstly, it is necessary to measure soil hydraulic properties to provide the model the information necessary to simulate soil moisture through the root zone. Secondly, it is recommended to have a weather station in the study to calculate the potential evapotranspiration demand. Unfortunately, some stakeholders have not the opportunity to have installed a weather station in the field, in this case, weather data must be downloaded from the nearest station. It is also highly recommended to install pressure head potential sensors to calibrate the model and verify that irrigation is triggered at the correct threshold."

Figure 3: The model underestimated soil water content at the 20 cm depth, which may lead to the overestimation of irrigation water.

During the field campaign one soil sample was collected at 10 cm depth. Soil hydraulic parameters are representative of this depth, that is why simulated soil moisture vales have a better agreement at 10 cm than 20 cm. Although, simulated soil moisture data at 20 cm depth is not exactly the same as field measurements, we consider that they are representative. One the one hand, the model reproduces the same soil moisture dynamics as field measurements. Thus, soil processes are well simulated. On the other hand, when irrigation is applied (from day 180th to day 225th) soil moisture data have a good agreement with sensors data. We specified in the manuscript that soil sample were taken at 10 cm depth (Line 226):

"One undisturbed soil sample was taken at 10 cm depth using a stainless-steel ring of 250 cm3 capacity."

Figure 6: The optimal scheduling requires to turn on and off the irrigation pump and system frequently, which may lead to increase in operation and maintenance costs. I

am wondering if such additional potential costs can be considered in the optimization framework.

Yes, it is considered in the optimization framework in Eq. 5 where Operational costs are defined. Note that the variable Ce is the energy cost. During the example exposed in this work, we did not have information about energy costs. For this reason, we did not calculate energy costs.

Table 4: RMSE values, 0.12 and 0.08 look substantial when considering the fact that the amount of available water content is around 0.35 from Figure 3. 0.12 and 0.08 correspond to 33% to 23%. Table 4: How about the overall bias?

Agreed. We found a mistake in Table 4 and we fixed it. The real RMSE at 10 cm depth corresponds to 0.012 and not 0.12. As we explained before, we took a soil core at 10 cm depth. Thus, we consider that the model simulates soil moisture at 20 cm correctly but the agreement between simulated and measured data is better at 10 cm depth.

Please also note the supplement to this comment:
https://hess.copernicus.org/preprints/hess-2020-146/hess-2020-146-AC1-supplement.pdf
* * *

---

## Author Comment (AC2) · 20 Jul 2020

Ref: hess-2020-146 Title: Combined Simulation and Optimization Framework for Irrigation Scheduling in Agriculture Fields Journal: Hydrology and Earth System Science

Response to Reviewer #2 Comments ####Authors' response in Blue####

Review on hess-2020-146 "Combined Simulation and Optimization Framework for Irrigation Scheduling in Agriculture Fields"

We are grateful to you for the time and effort spent on the review of our manuscript. Our detail response and comments raised by you is attached. We believe our responses

and the revisions made to the manuscript fully address the issues raised by the review. These revisions have helped clarify some aspects of our work and improve its interpretation.

GENERAL COMMENTS

The authors present an interesting and generally well-written manuscript about optimizing irrigation scheduling by combining 1-dimensional simulation of soil water movement and crop yield optimization. The research question is relevant. The methods used for simulation are well established. The results are interesting and promising for further research building upon the described methods.

SPECIFIC COMMENTS

Line 59: The publication of Hanson et al. (1977) might be too old as a reference to 'recent developments.

We changed the reference by (Line 60):

"Eigenberg, R. A., Doran, J. W., Nienaber, J. A., Ferguson, R. B. and Woodbury, B. L.: Electrical conducivity monitoring of soil condition and available n with animal manure and a cover crop, Agric. Ecosyst. Environ., 88, 183–193, doi:10.1016/S0167-8809(01)00256-0, 2002."

Line 74: You write "several authors" – are there more sources than Srivastava and Yeh (1991)?

We added two more references (Line 75):

Martínez-Gimeno, M. A., Jiménez-Bello, M. A., Lidón, A., Manzano, J., Badal, E., Pérez-Pérez, J. G., Bonet, L., Intrigliolo, D. S. and Esteban, A.: Mandarin irrigation scheduling by means of frequency domain reflectometry soil moisture monitoring, Agric. Water Manag., 235(March), 106151, doi:10.1016/j.agwat.2020.106151, 2020.

Jones, H. G.: Irrigation scheduling: Advantages and pitfalls of plant-based methods, J.

Exp. Bot., 55(407), 2427–2436, doi:10.1093/jxb/erh213, 2004.

Line 109: Please provide the units of qi at an earlier point than line 210.

We added the units [lÂům-2Âůh-1] (Line 114).

Line 147: The unit of h is not consistent with the rest of Richards' equation. Usually, the pressure head is given in units of length. Throughout the manuscript, please try to better distinguish 'pressure' from 'pressure heads.

We used pressure head denomination with hPa units because we followed the HYDRUS User Manual (Simunek and Sejna, 2014) indications. Based on our experience, soil pressure head in irrigation scheduling is usually expressed with hPa or kPa instead of units of length. That is the reason why we decided to express the pressure head in hPa.

Line 148: L is used as symbol for 'length unit' as well as for 'liter' (as in line 210, 287) in the manuscript. Please try to remove this ambiguity.

Thanks, we corrected this mistake and L is defined as length unit and l as liter volume of water units.

Line 256: Can you briefly indicate the method used for calibration?

We used a manual calibration. We added the information in the manuscript (Line 274).

Line 261: What is the distance between the study site and the nearest available weather station?

Thanks, we added the information and we modified the sentence as follows (Line 279):

"Meteorological parameters were downloaded from the nearest available weather station, located at 15 kms from the site, to…"

Line 335/336: 'Guarantee' is a strong word – maybe use a weaker one. I agree that the "optimal irrigation method" seems to ensure better conditions than the "traditional

method".

Thanks, we changed "guarantee" by "assumed" (Line 364).

Line 401: The word 'factor' is misleading – maybe better write "increase with respect to the traditional method by 7%". Moreover: Are 7% significant, considering the uncertainties involved? Agreed, we changed the sentence.

TECHNICAL COMMENTS Line 33: promote Agreed. (Line 33)

Line 34: designed Agreed. (Line 34)

Line 77: Campbell (1982) Agreed. (Line 78)

Line 83: stakeholders Agreed. (Line 86)

Line 84/85: capable of assessing Agreed. (Line 87)

Line 85: Siyal and Skaggs (2009) Agreed. (Line 87)

Line 90: later Agreed. (Line 92)

Line 90: irrigation scheduling Agreed. (Line 90)

Line 97: as follows Agreed. (Line 101)

Line 98: apply Agreed. (Line 102)

Line 99/100: capable of simulating Agreed. (Line 103)

Line 102: use Agreed. (Line 105)

Line 128: Units of Cy should be [EUR.t-1]. Agreed. (Line131)

Line 169: the models presented Agreed. (Line 173)

Line 289: constrained Agreed. (Line 315)

Line 328/329: capable of increasing Agreed. (Line 367)

Line 353: an increase Agreed. (Line 380)

Line 376: It should be Fig. 10. Agreed. (Line 404)

Line 381: It should be a panel of Fig. 10. Agreed. (Line 409)

Line 414: through the root zone instead of Agreed. (Line 448)

Line 482: Reference seems to be incomplete/broken. Agreed

Table 2: Usually, the unit of $\alpha$ is [m-1] or [cm-1]. Units of Ks should be [cm.d-1]. Please be consistent with the symbol for the third shape parameter i/I. Agreed

Table 4: Willmott Agreed

Table 5: For consistency, maybe better use 'L' as symbol for 'liter' Finally, we used L as length unit and l for liter unit.

Please also note the supplement to this comment:
https://hess.copernicus.org/preprints/hess-2020-146/hess-2020-146-AC2-supplement.pdf

────────────────────────────

---

## Author Comment (AC3) · 20 Jul 2020

Ref: hess-2020-146 Title: Combined Simulation and Optimization Framework for Irrigation Scheduling in Agriculture Fields Journal: Hydrology and Earth System Science

Response to Reviewer #3 Comments ####Authors' response in Blue####

Review on hess-2020-146 "Combined Simulation and Optimization Framework for Irrigation Scheduling in Agriculture Fields" We are grateful to you for the time and effort spent on the review of our manuscript. Our detail response and comments raised by you is attached. We believe our responses and the revisions made to the manuscript

fully address the issues raised by the review. These revisions have helped clarify some aspects of our work and improve its interpretation. The paper suggests a procedure for identifying the optimal irrigation schedule that maximize the net margin of the crop production. Irrigation scheduling is defined by a threshold of the soil matric potential observed at a given depth (e.g. 20 cm in the sample case study); ii) event irrigation depth (or irrigation duration with a predefined irrigation rate). The procedure exploits Hydrus-1D as soil water and solute transport model. Plan transpiration is modelled as fraction of the potential transpiration. The fraction is computed accounting for water and salinity stress. The potential transpiration and evaporation are computed as fractions of the crop evapotranspiration under standard conditions (ETc), accounting for the soil canopy cover. ETc is computed according to FAO-56 single crop coefficient approach. Crop yield is assumed to be proportional to the ratio of the actual ET to ETc.

GENERAL COMMENTS

Tabulated FAO 56 crop coefficients were proposed as a simple approach for assessing crop water requirements. The application of the single crop coefficient approach for estimating the crop evapotranspiration under standard conditions and the crop yield is too simplistic and not suited for the proposed optimization. By taking the tabulated crop coefficient in the proposed procedure is equivalent to assume that the crop is a stationary system, where phenology and water requirements are simply identified by the calendar days rather than the result of the crop response to the environmental conditions.

We agree with the reviewer in that estimating water requirements based on tabulated crop coefficient is too simplistic. There was a mistake in the manuscript and we did not specify that we used single crop coefficients. We used Kc values from Domínguez et al. (2012) and Martínez-Romero et al. (2017). They define maize Kc values calculated based on Growing Degree Days (GDD). The study area was located in Castilla la Mancha, where weather conditions are quite similar to that of Foradada. Thus, we assumed that these Kc values are good proxies for calculating water requirements.

We added this information in the manuscript (Line 280):

"K_c coefficients were extracted from the works of Domínguez et al.( 2012) and Martínez-Romero et al.( 2017), conducted in a maize field located in Castilla la Mancha, Spain. K_c coefficients were determined field temperature and an estimation of the Growing Degree Days (GDD). Weather conditions from Foradada field and Castilla la Mancha are similar. During the field campaigns, we visually corroborated the time duration of the different phenological stages proposed by Domínguez et al.( 2012) and Martínez-Romero et al.( 2017)."

We did not used the dual crop coefficient, who estimates evaporation and transpiration separately, because during the field campaign we estimated the canopy cover (Raes et al., 2010). The estimation of the soil fraction cover allows separating evaporation and transpiration from ETc. We assumed that it would be more realistic to apply the canopy cover based on field measurements than dual crop coefficient.

Similarly, Eq. 13 was proposed by FAO papers as a simple empirical equation for estimating crop yield. However, crop biomass and yield development depend on the transpiration rate rather than on the evapotranspiration.

We agree with the comment raised by the reviewer. Stewart (1977) model is a simple empirical model based on ETc values. Yet, it was proposed by FAO and has been widely used by several authors in the recent years (Domínguez et al., 2012; Irmak et al., 2016; Martínez-Romero et al., 2017; Saadi et al., 2015). In accordance with (Raes et al., 2010), we think that Stewart (1977) model is adequate for maize because these types of fields are totally covered most of the time We agree that transpiration is direclty related to crop yield, but in this case, soil evaporation is a minimum part of the total ETc. As we mentioned before, we used the canopy fraction cover for separating evaporation and transpiration. This canopy fraction cover, assumes that in systems with full canopy cover, such as maize, ETc is assumed to be equal to transpiration. In fact, Kool et al. (2014) reviewed the approaches for evapotranspiration partitioning and

validate this assumption.

We want to hihgly that the methodoogy does not impose Stewart (1977) model and allows to use another kind of crop productivity fuction. We clarify this in the mansucript as follows (Line 189):

"The model presented by Stewart et al. (1977) Eq. (13) has been widely accepted and recommended by the Food and Agriculture Organization of the United Nations (FAO) (Doorenbos and Pruitt, 1975, hereafter FAO24). In addition, it has been recently used by several authors (Domínguez et al., 2012; Irmak et al., 2016; Martínez-Romero et al., 2017; Saadi et al., 2015). Note though that methodology proposed is not limited by this model and can also be used with other soil – water – crop productivity models if needed."

Even the most simple and conceptual agro-hydrological model, such as AquaCrop (which does not rely on the numerical solution of the Richards Equations) provides a comprehensive description of the crop dynamics and crop yield development and, thus, allows optimizing the irrigation scheduling accounting for the crop response to environmental stresses.

The methodology proposed in this work aims to optimize the irrigation scheduling based on soil water content and pressure head status. For this, it is important use a model who simulates unsaturated water flux applying Richards equation, in this case HYDRUS. It is true that AquaCrop provides comprehensive description of the crop dynamics, but it provides approximated soil moisture data. Thus, AquaCrop does not follow the essence of our work, which is based on physical principles. As we have specified before, another kind of crop yield functions can also be used with the methodology proposed.

Indeed, an optimization procedure should consider that environmental stresses do not affect the yield uniformly across the entire growing cycle.

We agree with the comment raised by the reviewer. This is why Stewart (1977) model applies a crop response factor (Ky). This coefficient penalizes crop yields depending on crop phenology. For instance, during flowering stage, Ky = 1.05. It means that if actual evapotranspiration is not close to the potential water requirements (ETa/ETc ËĆ1) crop yield will be reduced at the end of the campaign.

Overall, it is not clear the motivation of this study. How should this procedure be applied from an operational perspective? The optimization procedure seems to be designed for running in batch mode, i.e. it can be used to identify the optimal irrigation schedule for a reference climatic condition, but it cannot be used to adapt the irrigation schedule to the actual environmental conditions, in real-time.

Thanks, this was not properly written in the paper. In principle, the combined simulation-optimization framework permits to find the optimal control settings of an irrigated field that maximize the net profit obtained in a period of time T given some forecasted climatic conditions. We have clearly state this now in Line 105. The method is not meant to update parameters in real-time based on new information. One should then incorporate a Bayesian statistical framework or similar which is not clearly included in the framework.

Line 99 "The framework permits to find the optimal control settings of an irrigated field that maximize the net profit obtained in a period of time T, given some forecasted climatic conditions. "

SPECIFIC COMMENTS Line 50 – The crop coefficient is designed for assessing crop water requirements and not for irrigation scheduling. The estimated crop water requirements should be then used for designing the irrigation scheduling.

We have modified the sentence to clarify this (Line 49):

"This method requires accurate estimations of weather conditions and does not provide the frequency and duration of irrigation (stakeholders do not know when to apply this

volume of water) unless water requirements are combined with soil moisture data."

Section 3.2 Model setup: Root depth is assumed to be constant in time, while it is highly variable in time, especially for crops like maize. A soil depth of 60 cm with free drainage as bottom boundary conditions does not seems to be realistic. Moreover, this seems even more improbable with crops like maize. The impact of the initial conditions can be high.

We did not describe this in the initial manuscript but we actually measured the root depth evolution during the field campaign. We introduced those measurements in the model. We have added this information in section 3.1, where we describe how and when we measured root depth (Line 237):

"In order to evaluate the vertical distribution of water uptake by plants we measured the root depth by pulling a plant off twice a month during the field campaigns. The maximum root depth registered was 55 cm after 78 days from sowing."

In section 3.2 we described the root growth model as follow (Line 266):

"Based on the root depth measurements taken during the field campaigns, we represent the vertical spatial distribution of water uptake by plants through Hoffman and Van Genucghten model (1983) with a root depth L_R of 55 cm, $\beta$(z)={âŰĹ(1.66667/L_R z>z_top-0.2L_R@2.0833/L_R (1-(z_top-z)/L_R ) zâĹĹ(z_top-L_R,z_top-0.2L_R)@0 z<z_top-L_R )âŤď'

We considered that it is appropriate consider 60 cm as a soil depth in the model, and also impose free drainage as a boundary condition. The reason is why we are interested in what happens in the root zone (how water moves thought the soil, roots water uptake, water evaporated...), but not what happens below this zone. Thus, considering that the maximum root depth was 55 cm, we assume that water percolates the root zone is lost by drainage. For this reason, we have to impose free drainage at the bottom of the soil profile.

Lines 325 – The irrigation strategy presented as traditional does not seem to be realistic

Based on our experience dealing with agricultures in Spain, we considered that irrigation scheduling based on water requirements can be simulated by adding the amount of water evapotranspired in the past during a certain period of time. We define that this period of time was a week instead of a day because it is the easier for agricultures to handle. If we define a new volume of water to apply every day (reschedule the irrigation every day), the agricultures will have to reprogram the irrigation controller every day, which is time consuming and not realistic.

Please also note the supplement to this comment:
https://hess.copernicus.org/preprints/hess-2020-146/hess-2020-146-AC3-supplement.pdf